# ARE LLMS AWARE THAT
# SOME QUESTIONS ARE NOT OPEN-ENDED?

## ABSTRACT

Large Language Models (LLMs) have shown the impressive capability of answering questions in a wide range of scenarios. However, when LLMs face different types of questions, it is worth exploring whether LLMs are aware that some questions have limited answers and have to respond more deterministically but some do not. We refer to the ability as *question awareness* that LLMs know to adjust the determinacy of the answers according to the questions. The lack of question awareness leads to two contradictory issues: (1) Too casual to answer non-open-ended questions. (2) Too boring to answer open-ended questions. In this paper, we first evaluate the question awareness ability of LLMs. The experimental results show that LLMs have the above issues of lacking the awareness of questions in certain domains, e.g. factual knowledge. To mitigate these issues, we propose a method called Question Awareness Temperature (QAT) sampling. This method enhances the question awareness ability of LLMs by dynamically adjusting the answer distributions based on question features. The automatic adjustment in QAT eliminates the need for manual temperature tuning in text generation. These findings underscore the potential of QAT sampling to enhance LLMs' question-awareness capabilities, thereby advancing their performance in various LLM benchmarks.

## 1 INTRODUCTION

Large language models (LLMs) (OpenAI, 2022; 2023; Anthropic, 2023) have emerged as groundbreaking innovations in the field of artificial intelligence, achieving a remarkable level of fluency and comprehension in question-answering using the human language (Taori et al., 2023; Chiang et al., 2023; Xu et al., 2023). Though LLMs can answer enormous questions with their knowledge base, we are hard to tell if the LLMs are aware of what kinds of questions they are answering. In other words, do LLMs understand that, open-ended questions encourage more casual and creative answers, but non-open-ended questions, e.g. problems about calculations and factual knowledge, need more accurate answers? We refer to this ability

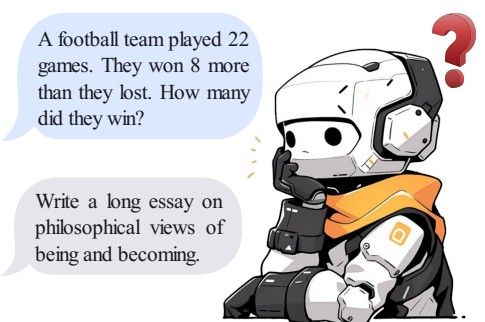

Figure 1: Are LLMs aware that some questions are not open-ended?

*question awareness* that one knows which type of questions requires deterministic answers and which does not.

The *question awareness* of LLMs indicates LLMs can identify which questions need more accurate answers and choose to act more deterministic. It is significant to explore that the question awareness of LLMs has a relationship to the model hallucinations and how to improve it because LLMs may be more likely to generate hallucinated answers when they are not sure.

In this paper, we first explore whether LLMs have the ability of question awareness across different types (open-ended/non-open-ended) of questions. Because LLMs sample the answer tokens from output distribution, we can examine the degree of the determinacy of LLMs from the steepness of

the output distributions. We can explore question awareness by checking if there is a difference in the output distribution when LLMs are asked different types of questions. We do the evaluation across different types of non-open-ended questions and open-ended questions. The experimental results show that LLMs have a certain degree of question awareness ability but lack the awareness of questions in some scenarios, e.g., factual knowledge, thus giving more casual and hallucinated answers in some cases.

To alleviate the influence of lacking question awareness ability, we propose Question Awareness Temperature (QAT) sampling, a method that enables LLMs to have question awareness to some extent and adjust the output distributions through temperature according to the different questions. In other words, when facing different questions, LLMs attain the ability to adaptively choose to be more deterministic or not by themselves avoiding the tedious process of temperature tuning.

To sum up, our contributions are stated as follows:

- We evaluate the question awareness ability of the LLMs and observe that LLMs have the fundamental ability to identify open-ended and non-open-ended questions but lack effective awareness of questions in some domains.

- We propose Question Awareness Temperature (QAT) sampling. It enables LLMs to choose to be deterministic or not when answering different questions by adaptively adjusting the sampling temperature.

- Our experimental results show that the QAT sampling enhances the question awareness ability of the LLMs and improves the performance on various LLM benchmarks.

## 2 PRELIMINARY EXPERIMENT: QUESTION AWARENESS EVALUATION

In this section, we evaluate the ability of question awareness that LLMs know which type of question needs to be answered more carefully in preliminary experiments. We first give a simple formulation of supervised fine-tuning (SFT) / instruction fine-tuning (Cobbe et al., 2021). Following the SFT formulation, we define the metric of question awareness using the steepness (kurtosis) of SFT output distributions as the indicator. Finally, we evaluate the question awareness ability on two widely used open-source LLMs, namely Llama 2 (Touvron et al., 2023) and Falcon (Penedo et al., 2023), across different question types. It is noted that we do not evaluate question awareness ability on GPT-3.5-turbo (OpenAI, 2022) and GPT-4 (OpenAI, 2023) because we can not obtain the output distributions from the APIs.

### 2.1 FORMULATION OF SUPERVISED FINE-TUNING

Supervised fine-tuning (SFT) serves as a bridge that leverages the foundational language comprehension gained during pre-training and then tailors it for general conversational purposes. For an auto-regressive language model, denoted as $\phi$, given a joint sequence $s = (x_1, x_2, \ldots, y_1, y_2, \ldots, y_T)$ of a question $x$ and an answer $y$ of length $T$, we minimize the SFT training objective only on the answer sequence $y$ in the teacher forcing fashion (Lamb et al., 2016):

$$\mathcal{L}_{SFT}(\phi) = \mathbb{E}\left(-\sum_{t=1}^{T} \log p_\phi(\hat{y}_t|x, y_{<t})\right). \tag{1}$$

During inference after supervised fine-tuning, we sample token from the output distribution $p_\phi(\hat{y}_t|x, y_{<t})$ to generate the token at step $t$.

### 2.2 METRIC

Following the formulation of supervived fine-tuning in Eq 1, the steepness of the output distribution $p_\phi(\hat{y}_t|x, y_{<t}) = (p_1, p_2, \ldots, p_n)$ indicates how LLMs are confident on the next token to predict among the entire token vocabulary with size $n$. A steeper distribution tends to be more like a one-hot vector, which means LLMs are more sure to predict the tokens with larger probabilities. We use the kurtosis to reflect how steep the distribution is. If the distribution is steeper, the kurtosis will get larger.

We use the average kurtosis of the distribution over the whole answer to reflect the determinacy of the answer. We calculate the average kurtosis $\mathcal{K}$ over the entire output distributions as follows:

$$\kappa_t = \frac{\frac{1}{n}\sum_{i=1}^{n}(p_i - \bar{p})^4}{\left(\frac{1}{n}\sum_{i=1}^{n}(p_i - \bar{p})^2\right)^2} - 3, \quad \mathcal{K} = \frac{1}{T}\sum_{t=1}^{T}(\kappa_t/\kappa_{\mathrm{one-hot}}), \tag{2}$$

where $p_i$ is the probability of the token to predict at step $t$ and $\kappa_t$ is the kurtosis of the distribution of the token at step $t$. We normalize the average kurtosis to $(0, 1)$ by dividing the kurtosis of the one-hot distribution $\kappa_{\mathrm{one-hot}}$.

## 2.3 EVALUATION PROCESS

In order to test the question awareness, we need to construct a question awareness evaluation dataset, where questions have distinctions in terms of the determinacy required to answer them. Therefore, we collect the questions of mainly two types, non-open-ended and open-ended questions. The non-open-ended questions have only fixed/limited answers: (1) **TRU**: commonsense knowledge that needs good understanding to answer, (2) **SMW**: diverse grade school math word problems, (3) **WDK**: memorization of world knowledge about histories, celebrities, places, books, movies, music, and etc. The open-ended questions are encouraged to have more creative answers about: (1) **CCR**: content creation including written articles, emails, and so on, (2) **DSC**: discussion on a certain topic, (3) **SUG**: suggestions offering. More details of this evaluation dataset are illustrated in Sec 4.3.

Using the evaluation dataset, we investigate two open-source LLMs with different sizes, including Llama 2-Chat 7b/13b/70b (Touvron et al., 2023), Falcon-instruct 7b/40b (Penedo et al., 2023). All of them are official chat versions, which have been fine-tuned for conversational usage in advance.

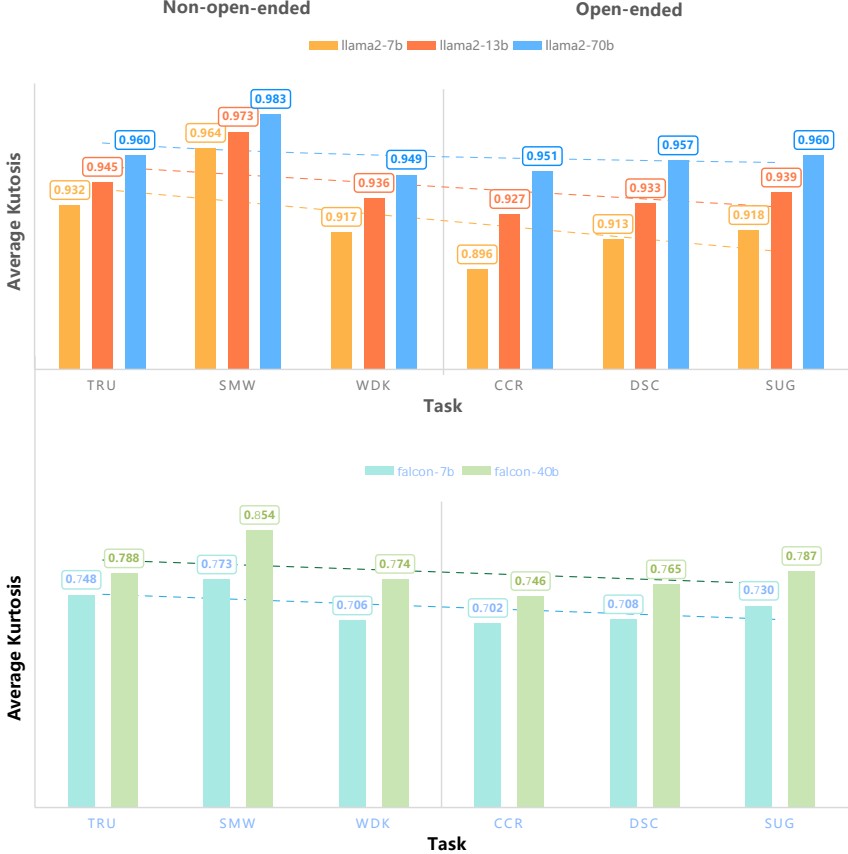

Figure 2: The result of question awareness evaluation of different LLMs. The figure above is of Llama 2 7b/13b/70b and the figure below is of Falcon 7b/40b. These dotted lines are the trend lines of the kurtosises, which are linearly fitted.

## 2.4 RESULTS AND ANALYSIS

**LLMs lack a strong sense of question awareness.** From the experimental results in Figure 2, we observe that the kurtosises of the non-open-ended questions are not significantly higher than the kurtosises of the open-ended questions in all models. For non-open-ended questions, though the LLMs have relatively high kurtosis in **SMW** and **TRU**, LLMs do not show more determinacy in the output distribution in the **WDK**, where the kurtosises are close to the average of open-ended questions. This shows that LLMs fail to recognize some questions about world knowledge are required to be answered carefully. For open-ended questions, similar problems can be found: Most LLMs have relatively lower kurtosis in **CCR**, but fail to be more creative and casual in **DSC** and **SUG**.

In conclusion, LLMs have fundamental question awareness on some scenarios, e.g. **SMW** and **CCR** but lack question awareness in some other scenarios, which means in some cases LLMs do not know when to choose to be more deterministic, thus leading casual and hallucinated answers to open-ended questions, and vice versa.

**Larger models have more confidence in text generation.** Though we do not observe an emergence of question awareness in larger models, we find that models with larger sizes tend to be more deterministic and focused in the generation. It means they are more confident in the generated answer and have higher accuracy.

## 3 QUESTION-AWARENESS TEMPERATURE SAMPLING

Based on the findings in the preliminary experiments about question awareness, we want to further improve the performance of LLMs by enhancing the question awareness ability of the LLMs in more scenarios. Therefore, we propose the Question-Awareness Temperature (QAT) Sampling, which adaptively adjusts the sampling temperature according to given questions. We first illustrate how sampling temperature affects the output distributions in text generation. Then we will introduce the mechanism of our QAT sampling step by step.

### 3.1 METHOD

The steepness of the output distribution can be adjusted by setting the temperature in the Softmax function as follows:

$$p_\phi(\hat{y}_t|x, y_{<t}) = \text{Softmax}\left(\frac{l_{\phi,t}(x, y_{<t})}{\mathcal{T}}\right), \quad \mathcal{T} \geq 0, \tag{3}$$

where the $l_{\phi,t}(x, y_{<t})$ is the output logit of the token at the step $t$. We can consider the naive Softmax function as a Softmax function with a temperature of 1. The lower the temperature is, the output distribution will get steeper and tend to be a one-hot vector, and vice versa. When the LLMs face different questions, we want the LLMs themselves to decide the determinacy of the answer by adjusting the temperature to change the steepness of output distributions. However, there is a challenge that temperature is a hyperparameter that needs tuning. Though we do not know the optimal temperature for generation, we can infer the tendency of how temperature changes according to the determinacy.

Therefore, in QAT sampling, we have two phases. In the first phase of continual fine-tuning, we first continue fine-tuning the LLMs of the chat version to predict how deterministic and focused they should be based on the given questions. In the second phase of inference with QAT sampling, we convert the predicted determinacy score to the sampling temperature and adaptively adjust the temperature on the fly during inference.

**Continual Fine-tuning Phase** We first continue fine-tuning the LLMs to have the ability to predict the determinacy. Therefore, we construct a dataset where questions are rated by a determinacy score. To be specific, we leverage the GPT-4 to rate the questions by the rules we make, which illustrate how deterministic the answers should be. For example, we let the GPT-4 rate the open-ended questions like **CCR** and **DSC** with lower scores and non-open-ended questions like **TRU** and **WDK** with higher scores. We use the questions as the input and the determinacy scores as the training labels.

We add a tiny network to the model to predict the determinacy score. This network consists of an attention layer and a small MLP with 1 hidden layer. The attention layer is initialized by the weights of the last layer of the original model. We collect the hidden states of the question sequence $x$, denoted as the $h_\phi(x)$. We feed the $h_\phi(x)$ to the attention layer to get a new hidden states $h'_\phi(x)$. We use the average pooling to get a global feature vector and feed the vector into the MLP. The network tries to predict the target $\tau$ by minimizing the Mean Square Error (MSE) loss as follows:

$$\hat{\tau} = \text{Sigmoid}(\text{MLP}(\text{AvgPool}((h'_\phi(x))), \quad h'_\phi(x) = \text{Attn}(h_\phi(x)), \tag{4}$$

$$\mathcal{L}_{QAT}(\phi) = \frac{1}{2}(\tau - \hat{\tau})^2. \tag{5}$$

During the continual fine-tuning, we freeze the weights of the model and only fine-tune the weights of the additional network, avoiding the degradation in the original SFT performance.

Besides that, we need to record the mean and standard deviation of the kurtosis of the output distributions during training, denoted as $\mathcal{K}_{avg}$ and $\mathcal{K}_{std}$. We record these values for the inference phase later. We calculate the $\mathcal{K}_{avg}$ and $\mathcal{K}_{std}$ using the exponential moving average as follows:

$$\begin{aligned} \mathcal{K}_{avg,s} &= \beta \cdot \mathcal{K}_{avg,s-1} + (1 - \beta) \cdot \hat{\mathcal{K}}_{avg,s}, \\ \mathcal{K}_{std,s} &= \beta \cdot \mathcal{K}_{std,s-1} + (1 - \beta) \cdot \hat{\mathcal{K}}_{std,s}, \end{aligned} \tag{6}$$

where the $\hat{\mathcal{K}}_{avg,s}$ and $\hat{\mathcal{K}}_{std,s}$ are calculated by averaging the means and standard deviations of kurtosis of the whole batch at training step $s$.

**Inference Phase**  In QAT sampling, before starting the inference, we will first use LLMs to predict the determinacy score $\hat{\tau}$ from the hidden states of the question tokens $x$ in Eq 4. If the determinacy score is close to 1 (getting higher), it means the LLMs are required to be more deterministic to answer the question. The prediction of the determinacy score will be done only once at the start of the generation.

Though we can simply rescale the determinacy score to get the temperature, it is noted that predicting the temperature in this way does not take into account the output distributions of the generated answer. From the findings in question awareness evaluation in Sec 2, we observe that LLMs have the fundamental question awareness ability in some cases, which means some output distributions are steep/flat enough to give a deterministic/creative answer. What we need to do is to revise the unawareness cases and avoid overcorrection. Thus we need to take into account the question awareness ability which is inherent to the LLMs themselves.

In order to avoid overcorrection, we transfer the original kurtosis of the output distributions to a target kurtosis we set. The target kurtosis takes the value from the kurtosis interval $[\mathcal{K}_{lower}, \mathcal{K}_{upper}]$, which is calculated below:

$$\begin{aligned} \mathcal{K}_{upper} &= \mathcal{K}_{avg} + \lambda \cdot \mathcal{K}_{std}, \\ \mathcal{K}_{lower} &= \mathcal{K}_{avg} - \lambda \cdot \mathcal{K}_{std}, \end{aligned} \tag{7}$$

where the $\mathcal{K}_{avg}$ and $\mathcal{K}_{std}$ are recorded in Eq 6 during the continual fine-tuning phase. The kurtosis interval represents the range that normally kurtosis of the model output distribution can reach. According to the kurtosis interval, we use the predicted determinacy score $\hat{\tau}$ to calculate a target kurtosis $\mathcal{K}_{target}$ proportionately from the interval as follows:

$$\mathcal{K}_{target} = \hat{\tau} \cdot (\mathcal{K}_{upper} - \mathcal{K}_{lower}) + \mathcal{K}_{lower}, \quad 0 \le \hat{\tau} \le 1, \tag{8}$$

The target kurtosis $\mathcal{K}_{target}$ lies in the kurtosis interval, which constrains the range of the kurtosis of adjusted output distributions. It avoids overcorrection that the adjusted distributions are too steep or too flat. Therefore, next our goal is to adjust the output distributions thus we can transfer the 'original kurtosis' to the target kurtosis. Before that, we use the mean of the kurtosis of the generated token distributions to represent the 'original kurtosis':

$$\kappa_{avg,t} = \frac{1}{t} \sum_{i=1}^{t} \kappa_i, \tag{9}$$

The $\kappa_{avg,t}$ is a running mean which is updated as the number of generated tokens increases. It can be inferred that as the step $t$ increases, the running mean $\kappa_{avg,t}$ will be approximate to the true average kurtosis of the output distributions. By changing the temperature of the Softmax function, we can adjust the temperature $\hat{\mathcal{T}}_t$ to transfer the average kurtosis $\kappa_{avg,t}$ to approximate the target kurtosis $\mathcal{K}_{target}$:

$$\hat{\mathcal{T}}_t = 1 + \eta \cdot (\kappa_{avg,t} - \mathcal{K}_{target})t, \tag{10}$$

$$\hat{\mathcal{T}}_t = \text{Clamp}(\hat{\mathcal{T}}_t, \mathcal{T}_{min}, \mathcal{T}_{max}). \tag{11}$$

In Eq 10, the adjustment of temperature is decided by three factors: (1) the difference between the $\kappa_{avg,t}$ and $\mathcal{K}_{target}$, (2) the number $t$ of generated tokens, (3) a coefficient $\eta$ to control the adjustment speed. For the first factor, if $\kappa_{avg,t} > \mathcal{K}_{target}$, it means the kurtosis of output distributions is higher than the target kurtosis, thus we need to increase the temperature to flatten the output distributions, and vice versa. For the second factor, as the number of generated tokens increases, the $\kappa_{avg,t}$ tends to approach the true average kurtosis of the output distributions. Thus as the $t$ increases, the $(\kappa_{avg,t} - \mathcal{K}_{target})$ exerts a greater impact on the temperature adjustment. Finally, we need to clamp the temperature between an interval to avoid being too high or too low in Eq 11.

To sum up QAT sampling, we realize it in two phases. In the continual fine-tuning phase, we construct a training dataset to train a determinacy score predictor. In the inference phase as shown in Algorithm 1, we adjust the temperature mainly according to the difference between current average kurtosis and target kurtosis. Essentially, QAT sampling does not predict the temperature but uses the temperature to revise the steepness of output distributions to where it should be while in the constraint of the kurtosis interval. The key difference is that simply changing the temperature may lead to overcorrection ignoring the LLM inherent question awareness ability. By contrast, QAT sampling adjusts the temperature based on the original kurtosis interval to avoid these situations.

---

**Algorithm 1** QAT Sampling in the inference phase

---

**Input:** hiddent states $h_\phi(x)$ of question $x$, output logits $l_{\phi,t}(x, y_{<t})$, kurtosis mean $\mathcal{K}_{avg}$ and kurtosis standard deviation $\mathcal{K}_{std}$ recorded in the continual fine-tuning phase.

**Output:** answer sequence $y$

$h'_\phi(x) = \text{Attn}(h_\phi(x))$

$\hat{\tau} = \text{Sigmoid}(\text{MLP}(\text{AvgPool}(h'_\phi(x)))) \qquad \triangleright$ Determinacy score prediction in Eq 4

$\mathcal{K}_{upper} = \mathcal{K}_{avg} + \lambda \cdot \mathcal{K}_{std}, \mathcal{K}_{lower} = \mathcal{K}_{avg} - \lambda \cdot \mathcal{K}_{std} \qquad \triangleright$ Kurtosis interval in Eq 7

$\mathcal{K}_{target} = \hat{\tau} \cdot (\mathcal{K}_{upper} - \mathcal{K}_{lower}) + \mathcal{K}_{lower} \qquad \triangleright$ Target kurtosis in Eq 8

$t = 1, \mathcal{T}_0 = 1.0, y = [\,] \qquad \triangleright$ Initialization

**repeat**

$\quad \hat{p_\phi}(\hat{y}_t|x, y_{<t}) = \text{Softmax}\left(\frac{l_{\phi,t}(x, y_{<t})}{\mathcal{T}_{t-1}}\right)$

$\quad \kappa_t = \frac{\frac{1}{n}\sum_{i=1}^{n}(p_i - \bar{p})^4}{\left(\frac{1}{n}\sum_{i=1}^{n}(p_i - \bar{p})^2\right)^2} - 3 \qquad \triangleright$ Calculate kurtosis over $\hat{p_\phi}(\hat{y}_t|x, y_{<t})$

$\quad \kappa_{avg,t} = \frac{1}{t}\sum_{i=1}^{t}\kappa_i$

$\quad \hat{\mathcal{T}}_t = 1 + \eta \cdot (\kappa_{avg,t} - \mathcal{K}_{target})t \qquad \triangleright$ Adjust the temperature in Eq 10

$\quad \hat{\mathcal{T}}_t = \text{Clamp}(\hat{\mathcal{T}}_t, \mathcal{T}_{min}, \mathcal{T}_{max})$

$\quad p_\phi(\hat{y}_t|x, y_{<t}) = \text{Softmax}\left(\frac{l_{\phi,t}(x, y_{<t})}{\mathcal{T}_t}\right)$

$\quad \hat{y}_t = \text{Sample}(p_\phi(\hat{y}_t|x, y_{<t})) \qquad \triangleright$ Sample next token from adjusted output distribution

$\quad y = \text{Append}(y, \hat{y}_t) \qquad \triangleright$ Append new generated token to answer

$\quad t = t + 1$

**until** $\hat{y}_t == <|\text{endoftext}|> \qquad \triangleright$ Exit when meeting the eos token

**return** $y$

---

## 4 EXPERIMENTS

### 4.1 CONTINUAL FINE-TUNING

#### 4.1.1 QAT TRAINING DATASET

We carefully filter 3.5k high-quality dialogues with different question types from ShareGPT dataset(Dom Eccleston, 2023). We leverage the GPT-4 to label the questions with the determinacy scores according to how deterministic the answers should be. We rate the questions for 4 levels from most creative (level 1) to most deterministic (level 4). We let GPT-4 label each question twice and average the two ratings as the final level, as shown in Figure 4. We rescale the level score to (0, 1) as the final determinacy score. The detailed rating criteria and how we prompt GPT-4 to label the data can be checked in Appendix A.

#### 4.1.2 TRAINING DETAILS

We continue fine-tuning two widely used LLMs of different sizes, namely Llama 2-Chat 7b/13b/70b (Touvron et al., 2023), Falcon-instruct 7b/40b (Penedo et al., 2023). All these models have been supervisedly fine-tuned in advance by the official publishers. We continue fine-tuning these LLMs on our training dataset. We observe that the QAT loss $\mathcal{L}_{QAT}$ quickly converges in approximately 100 steps thus we train for only 1 epoch on the training dataset with a global batch size of 32. We train the 7b models with a learning rate of 2e-5, the 13b model with 1e-5, and the 40b/70b models with 5e-6.

### 4.2 EVALUATION

In order to verify the effectiveness of the QAT sampling. We use our question awareness evaluation dataset in Section 2 to evaluate if the LLMs with QAT sampling have a better awareness of different question types and have better performance than the LLMs with naive temperature sampling. Besides that, we choose two comprehensive LLM benchmarks, namely AlpacaEval (Li et al., 2023) and MT-Bench (Zheng et al., 2023), to verify the general performance of the LLMs with QAT sampling. We set the original LLMs using the naive temperature sampling of temperature 1.0 as the baselines.

### 4.3 EVALUATION DATASETS

**Question Awareness Evaluation Dataset** The question awareness evaluation dataset consists of two types of questions, open-ended and non-open-ended questions. This dataset can also be used to evaluate the performance of LLMs on different question types except for the evaluation of question awareness ability in Sec 2.

The open-ended type has three subsets: (1) **TRU**: We select 100 hard questions about commonsense knowledge from the TruthfulQA dataset (Lin et al., 2022). (2) **SMW**: We select 100 school math word problems of diverse grades from the GSM8k dataset (Cobbe et al., 2021), where each question needs a certain amount of reasoning besides calculations. (3) **WDK**: We select 100 questions about word knowledge from RefGPT-Fact dataset (Yang et al., 2023). RefGPT-Fact is a multi-turn dialogue dataset adaptively constructed from Wikipedia, including knowledge of histories, celebrities, places and so on. The non-open-ended type also has three subsets: (1) **CCR**: content creation like writing something according to the human's instruction, (2) **DSC**: discussing or being asked for perspectives on a certain topic, (3) **SUG**: offering some useful suggestions including traveling, studying, and so on. All these subsets of non-open-ended type have 100 questions each and are carefully selected by humans from ShareGPT datasets (Dom Eccleston, 2023).

**AlpacaEval** The AlpacaEval (Li et al., 2023) evaluation set has about 800 single-turn questions covering a wide range of scenarios to evaluate chat assistants.

**MT-Bench** MT-bench Zheng et al. (2023) contains a set of 80 challenging multi-turn questions for evaluating chat assistants, including writing, roleplay, reasoning, math, coding, and so on.

## 4.4 EVALUATION PROCESS

Considering the huge amount of work of checking whether the model's answers are correct or not, we leverage the GPT-4 as the automatic evaluator, where the effectiveness has been verified in AlpacaEval and MT-Bench. For the question awareness dataset, **TRU** and **SMW** both have unique standard answers, we use the accuracy to measure the performance of these two tasks. **WDK** about world knowledge has the references for GPT-4 to check if the model's answers contain hallucination. We let GPT-4 score the answers according to the degree of hallucination in **WDK**. Other three open-ended tasks of **CCR**, **DSC** and **SUG** do not have standard answers, thus we use the judgment prompt from MT-bench to score the answers according to the criteria in the prompt. For the AlpacaEval and MT-bench, similar to the open-ended tasks, the models' answers are scored by GPT-4 using the MT-bench judgment prompt.

## 5 RESULTS AND ANALYSIS

From Figure 3, we evaluate the question awareness ability of the Llama 2 models using QAT sampling. The descending trend lines have shown a distinction in the awareness between the non-open-ended questions and open-ended questions. The models with QAT sampling choose to be more deterministic in answering the non-open-ended questions, thus we can observe higher kurtosises in non-open-ended tasks. Similar findings can be seen in open-ended questions.

In table 1, we can see that QAT sampling largely improves the LLM performance in the various tasks, especially in the non-open-ended questions. It means that a better awareness of non-open-ended questions can reduce the hallucination to some extent.

For the results of two comprehensive LLM benchmarks, both Llama 2 and Falcon have significant improvements over the baselines, which shows the QAT sampling is useful for different models with different sizes. We observe that smaller models like Llama 7b and Falcon 7b have more performance gains than larger models. It can be inferred that the distribution of larger models originally has more appropriate tokens with high probabilities thus the effectiveness of additional adjustment on the steepness of the distribution tends to be smaller.

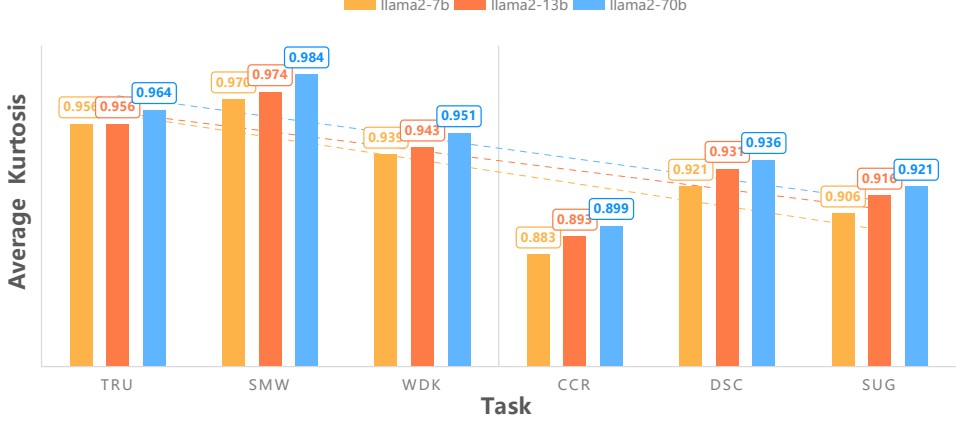

Figure 3: The result of question awareness evaluation of different LLMs using the QAT sampling.

## 6 RELATED WORK

The methods of controlling text generation in LLMs have seen significant advancements in recent years. Sampling methods play a crucial role in controlling the output quality, diversity, and creativity of generated text. This section provides an overview of the key techniques related to sampling in text generation with LLMs. The development in this area can be broadly categorized into three main themes: 1) Naive Sampling Strategies, 2) Temperature Scaling, and 3) Post-processing Techniques.

Table 1: Evaluating the performance of LLMs using QAT sampling on various LLM benchmarks. Acc represents the accuracy and Sco represents the score (1 to 10).

| Model | TRU Acc | SMW Acc | WDK Sco | CCR Sco | DSC Sco | SUG Sco | AlpacaEval Sco | MT-Bench Sco |
|---|---|---|---|---|---|---|---|---|
| **Llama 2 7b** | 50.0 | 21.0 | 5.15 | 9.19 | 9.35 | 9.40 | 8.51 | 6.88 |
| + QAT | 55.0 | 29.0 | 5.61 | 9.07 | 9.35 | 9.43 | 8.71 | 7.19 |
| **Llama 2 13b** | 62.0 | 43.0 | 5.78 | 9.22 | 9.27 | 9.45 | 8.81 | 7.35 |
| + QAT | 63.0 | 46.0 | 6.07 | 9.25 | 9.30 | 9.51 | 8.96 | 7.56 |
| **Llama 2 70b** | 59.0 | 62.0 | 6.65 | 9.33 | 9.48 | 9.49 | 9.20 | 7.78 |
| + QAT | 61.0 | 64.0 | 6.85 | 9.29 | 9.50 | 9.47 | 9.24 | 7.83 |
| **Falcon 7b** | 26.0 | 2.0 | 2.79 | 6.21 | 6.28 | 6.61 | 5.45 | 4.50 |
| + QAT | 32.0 | 2.0 | 3.28 | 6.41 | 6.58 | 6.72 | 5.82 | 5.11 |
| **Falcon 40b** | 50.0 | 13.0 | 4.66 | 7.33 | 7.91 | 8.21 | 7.26 | 6.30 |
| + QAT | 53.0 | 15.0 | 4.98 | 7.57 | 8.01 | 8.16 | 7.42 | 6.59 |

**Naive Sampling Strategies**   Greedy sampling selects the token with the highest predicted probability at each step, resulting in deterministic and often repetitive text. Beam search (Freitag & Al-Onaizan, 2017) maintains a beam of top-k candidates at each step, expanding multiple possibilities in parallel. It improves diversity compared to greedy sampling but may still produce repetitive or ungrammatical text. Random sampling selects tokens based on their predicted probabilities, introducing randomness into the generation process. It enhances diversity but can produce inconsistent or nonsensical text.

**Temperature Scaling**   The naive sampling strategies only consider the original output distributions. The temperature in the Softmax function can control the distribution of token probabilities. Temperature scaling can be adjusted to tune the trade-off between creativity and coherence in the generated text. In this paper, the QAT sampling can adaptively adjust the temperature to control the steepness of output distributions.

**Post-processing Techniques**   Because the tokens with higher probabilities are probably appropriate choices, we can choose only to select these tokens, avoiding sampling nonsensical tokens. Top-k sampling (Fan et al., 2018) narrows down the token selection to the top-k most probable tokens, increasing the likelihood of coherent text and balancing diversity and quality. Similar to the motivation of top-k sampling, nucleus sampling (Holtzman et al., 2020), also known as top-p sampling, dynamically selects the top-p fraction of tokens with the highest probabilities. Locally typical sampling (Meister et al., 2023) posits the abstraction of natural language generation as a discrete stochastic process and samples tokens according to conditional entropy. Entmax sampling (Martins et al., 2020) leverages entmax transformation to train and sample from a natively sparse language model. Keyword-based sampling (au2 & Akhtar, 2023) uses knowledge distillation techniques to extract keywords and samples using these extracted keywords. It is noted that these post-processing techniques are capable with the QAT sampling, as QAT sampling directly adjusts the output distribution of LLMs.

## 7   CONCLUSION

In this paper, we highlight the question awareness ability of LLMs, which receives little attention from previous studies. While LLMs exhibit a fundamental awareness of open-ended and non-open-ended questions, they do falter in certain domains, often leading to casual or inaccurate responses. To bridge the gap, we introduce Question Awareness Temperature (QAT) sampling, enabling LLMs to autonomously adapt their response determinacy based on question type. Our experiments showcased the efficacy of QAT, significantly enhancing LLM performance across various benchmarks.

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

# A  QAT TRAINING DATASET

The QAT training dataset consists of 3.5k high-quality questions, which are carefully filtered from the ShareGPT dataset (Dom Eccleston, 2023). We leverage the GPT-4 to label the questions with the determinacy scores according to how deterministic the answers should be. The prompt and our criteria are shown in Table 2. We rate the questions for 4 levels from most creative (level 1) to most deterministic (level 4), as shown in Figure 4. We rescale the level score to (0, 1) as the final determinacy score.

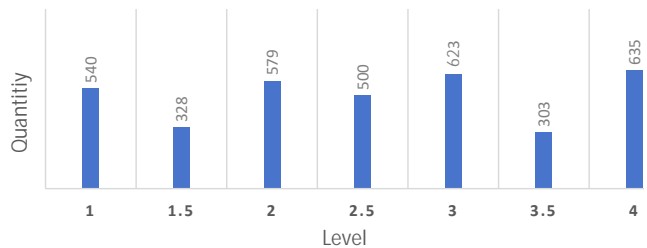

Figure 4: The allocation of each level in the QAT training dataset.

Table 2: The prompt for collecting QAT training dataset.

---

**System prompt:**
You need to assess which user inquiries/instructions require a more precise response from the AI assistant based on the Judgment Criteria and assign scores according to the Judgment Criteria.

**User prompt:**
Judgment Criteria:

- Highly Accurate (4 points): Questions/instructions that have a unique answer, including mathematical calculations, factual knowledge inquiries (such as history, medicine, biology, physics, astronomy, law, geography, economics, chemistry, general knowledge, etc.). These require the most precise responses.

- Fairly Accurate (3 points): Questions/instructions related to logical reasoning, code modification and creation, text rewriting and summarization, text translation, reading comprehension (answering questions based on provided text), etc. These have relatively consistent answers but might involve some interpretation.

- Moderately Accurate (2 points): Questions/instructions related to code discussions and creative inquiries that require a certain level of expertise. These involve creativity but still require accuracy to some extent.

- Not Very Accurate (1 points): Creative and open-ended questions/instructions (e.g., "What do you think about...?" "How do you see...?"). These do not require precision but emphasize creativity and viewpoints.

According to the judgment criteria, considering the type of questions/instructions from the user, the AI assistant's responses need varying degrees of accuracy. Questions/instructions with multiple possible answers and lacking a single correct response require lower precision and receive lower scores. Provide ratings for the question (1-4 points) in the format: "Question: x points". Explain first and then present the score.

---

