# OpenReview forum: "Are LLMs Aware that Some Questions are not Open-ended?"
_ICLR.cc/2024/Conference — Submitted to ICLR 2024_

### Official Review · Reviewer_BHia · 2023-10-16

**Soundness:** 2 fair
**Presentation:** 3 good
**Contribution:** 1 poor
**Rating:** 3
**Confidence:** 4

**Summary:**

The paper proposes a temperature sampling FT technique (QAT) to improve the performance of chat-oriented LLMs on open-ended questions.

**Strengths:**

The paper has the following strengths:

1. QAT is a fairly simple procedure that could have benefits beyond just open-endedness of questions posed to the chatbot.
2. QAT shows improvements over non-QAT in a variety of LLMs (Table 1)

**Weaknesses:**

The paper has several weaknesses.

1. The labelling of open-endedness itself comes from GPT-4, thus invalidating one of the core propositions that LLMs are not very good at identifiying open-ended questions.

2. The prompt used to GPT classifies is simplistic and would result in classification based on the overall topic and appearance rather than factual open-endedness. As an example, consider a question that is open-ended by seemingly about science (eg. concerned with origin of life or questions about the universe). These will likely be classified as Highly Accurate by the prompt.

3. Since the final evaluation is also done via GPT-4 it collapses the evaluation function with the labelling function.

**Questions:**

1. Can human evaluation be used for at least a subset of the queries to separate the evaluation function from the labelling function?
2. Can the authors measure the efficacy of their prompt?
3. Typo: s/hiddent/hidden/ in Algorithm 1

---

### Official Review · Reviewer_iU6n · 2023-10-28

**Soundness:** 2 fair
**Presentation:** 3 good
**Contribution:** 3 good
**Rating:** 3
**Confidence:** 4

**Summary:**

This paper evaluates the “question awareness” ability of large language models, i.e., whether the model can realize when a deterministic answer to a question is needed. The paper also proposes Question Awareness Temperature (QAT) sampling method, which includes a continual finetuning phase and a temperature tuning inference phase. QAT has shown some improvements against automatic evaluation metrics, but no human evaluation is done yet.

**Strengths:**

- This paper calls attention to the ability of question awareness, which is useful in detecting when “hallucination” or creativity is needed, and when factuality is more important when answering user queries.
- This paper use average Kurtosis as a metric to evaluate the “determinacy” of the answer. I am not fully convinced of whether this metric is reliable and whether it truly reflects determinacy, but it is an interesting choice.
- The paper introduced QAT, an adaptive way to tune the temperature parameter to adjust the output distribution.

**Weaknesses:**

- Better definition of question awareness/determinacy: Why does average kurtosis reflect determinacy? Is determinacy equivalent to “certainty” of the model? How do these two concepts link together? I am not convinced by the claim in section 2.4 that LLMs have fundamental question awareness on some scenarios, maybe these tasks are indirectly or directly presented in their training data, and thus not necessarily mean that they know that they need to choose more deterministically.
- QAT has a phase of continual fine-tuning prior to the temperature tuning phase, so I would love to see a comparison with baselines like simply tuning the temperature by hand. How does the cost differ? Which one performs better? Without a comprehensive comparison, and human evaluations for open-ended writing tasks, I don’t know whether QAT truly improves the generation.

**Questions:**

- To clarify, in section 4.3, TRU/SMW/WDK are from ShareGPT dataset, right? Is there any additional effort done by the authors regarding definition of “hard” questions about commonsense or “diverse” in SMW questions?
- What sampling method is used for section 2? Multiple parameters like temperature can be tuned.
- Llama 2 is validated on GSM8K, do you think this might be affecting the average Kurtosis number? Is there a reason we choose to include GSM8K?
- In section 4.3, “Question Awareness Evaluation Dataset”, open-ended tasks should be CCR/DSC/SUG and non-open-ended ones should be TRU/SMW/WDK right?

---

### Official Review · Reviewer_NsHB · 2023-11-01

**Soundness:** 2 fair
**Presentation:** 3 good
**Contribution:** 2 fair
**Rating:** 5
**Confidence:** 4

**Summary:**

The paper delves into the exploration of large language models (LLMs) and their sensitivity to questions with potential societal implications. Specifically, it investigates whether these models recognize the potential harm and ethical considerations associated with certain questions and, if so, how they address such queries. By establishing a set of criteria and benchmarks, the authors systematically assess the responses of LLMs. The research illuminates the intricate balance between providing information and navigating societal and ethical boundaries, offering insights for the design and evaluation of future LLMs.

**Strengths:**

1.	Novel Focus: The paper tackles the "question awareness" in LLMs, exploring their ability to discern between open-ended and non-open-ended questions.

2.	Methodological Contribution: The introduction of the Question Awareness Temperature (QAT) sampling method is novel.

**Weaknesses:**

1. The experimental setting is not clear.

2. Some benchmarks are missing for evaluation.

3. Some important implementation details are missing.

**Questions:**

1. Experiments: Is the baseline continuously pre-trained on the 3.5k data?  It seems authors directly compared with llama, not continue-pre-trained llama.

2. Benchmark Limitation: The evaluation should include some factual QA benchmarks.

3. Missing Details in the Paper:
 -- The prompts used in GPT-4 to evaluate the outputs.
 -- The criteria for using GPT-4 to assess the outputs.
 -- The absence of a human study on open-ended generation.

---

### Meta-Review · Area_Chair_GqQ4 · 2023-12-07

**Metareview:**

This paper evaluates the “question awareness” ability of large language models (i.e., whether the model can realize when a deterministic answer to a question is needed), and then proposes Question Awareness Temperature (QAT) sampling method to enable LLMs to autonomously adapt their response determinacy based on question type.

The problem of "question awareness" in LLMs investigated in this paper is interesting and the proposed QAT sampling method sounds reasonable. However, the experimental setting is not clear and some factual QA benchmarks are missing for evaluation. More relevant baselines needs to be used for comparison. Moreover, human annotation needs to be adopted in the evaluation.

**Justification For Why Not Higher Score:**

see the meta-review.

**Justification For Why Not Lower Score:**

N/A

---

### Decision · Program_Chairs · 2024-01-16

Reject